# Sequence Metric Learning as Synchronization of Recurrent Neural Networks

## Abstract

Sequence metric learning is becoming a widely adopted approach for various applications dealing with sequential multi-variate data such as activity recognition or natural language processing and is most of the time tackled with sequence alignment approaches or representation learning. In this paper, we propose to study this subject from the point of view of dynamical system theory by drawing the analogy between synchronized trajectories produced by dynamical systems and the distance between similar sequences processed by a siamese recurrent neural network. Indeed, a siamese recurrent network comprises two identical sub-networks, two identical dynamical systems which can theoretically achieve complete synchronization if a coupling is introduced between them. We therefore propose a new neural network model that implements this coupling with a new gate integrated into the classical Gated Recurrent Unit architecture. This model is thus able to simultaneously learn a similarity metric and the synchronization of unaligned multi-variate sequences in a weakly supervised way. Our experiments show that introducing such a coupling improves the performance of the siamese Gated Recurrent Unit architecture on an activity recognition dataset.

## 1 Introduction

Metric learning aims at learning an essential component for numerous machine learning algorithms used for classification or clustering: a similarity. It has the benefit to be usable in weakly supervised settings where only equivalence constraints between samples are known (Xing et al. (2003)), which allows for a large number of applications on various data types: from person re-identification (Yang et al. (2018)), object tracking (Bertinetto et al. (2016)) and gesture recognition (Berlemont et al. (2018)) to sentence similarity computation (Mueller & Thyagarajan (2016)). Among those applications, less attention has been given to design specific sequence metric learning algorithms, specifically with neural networks despite the simplicity of the siamese architecture (Bromley et al. (1994)).

One easy way to adapt existing approaches to sequential data is to learn representations through Sequence-to-Sequence models (Sutskever et al. (2014)) or Transformers (Vaswani et al. (2017)). However, these models would be difficult to learn in a weakly supervised way for providing a similarity metric and further lose temporal dependency information inside the sequence and alignment information between sequences. On the contrary, Dynamic Time Warping (DTW) (Sakoe & Chiba (1978)) is a classical approach to measure distance between sequences and relies on aligning sequences. Its integration inside learning algorithms has been rendered difficult by its non-differentiability and its theoretical quadratic time complexity which badly suits the equivalence constraint framework and some associated more complex losses (Oh Song et al., 2016; Sohn, 2016; Yang et al., 2018). Recent works mitigate these drawbacks notably with virtual metric learning (Perrot & Habrard, 2015; Su & Wu, 2019) and soft versions of DTW (Cai et al., 2019; Abid & Zou, 2018).

Therefore, we aim at designing a neural network architecture specifically adapted to sequence metric learning. Recurrent neural networks (RNN) have a temporal dynamic behavior which allows to study them as dynamical systems. We propose in this paper a new framework for sequence metric

learning based on dynamical system synchronization theory. We propose to replace the concept of metric in a vector space by the concept of synchronization of trajectories in a state space. Instead of computing distances on input representations, we propose to measure how two dynamical systems, and precisely two RNNs, respond to input pairs in term of synchronization. The notion of coupling is crucial when trying to synchronize dynamical systems. We introduce a coupled version of the Gated Recurrent Unit (GRU) (Cho et al. (2014)) to implement coupling inside a siamese architecture. Our experimental evaluation shows that this modification provides an improvement over a classical Siamese GRU implementation.

The paper is organized as follows: Section 2 outlines the state-of-the-art approaches in sequence metric learning, Section 3 describes our framework and our new siamese architecture, Section 4 shows our experimental results to assess the performances of our approach compared to the state of the art, and Section 5 presents our conclusions and perspectives.

## 2 RELATED WORK

**Recurrent neural networks and dynamical system theory.** A main property of RNN is to exhibit a dynamic behavior which enables them to learn temporal sequence correlations. This behavior can therefore be studied using dynamical system theory: an important result being that RNN can approximate any finite-time trajectory of a dynamical system (Funahashi & Nakamura (1993)). Other early works analyzed the RNN convergence stability (Hirsch (1989)) and helped to understand the problem of long term dependencies (Bengio et al. (1994)). Laurent & von Brecht (2016) studied the dynamics of Long-Short Term Memory (LSTM) Neural Networks and GRU and observed that it is chaotic in the absence of input data. They designed a Chaos-Free RNN architecture having a more predictable behavior. In another recent publication, Chang et al. (2019) studied RNN trainability and established a connection with discretized Ordinary Differential Equations stability. They identified a criterion to guarantee that the system can preserve long-term dependencies and proposed a new version of RNN based on those observations. Both papers demonstrate that dynamical system theory is a fertile soil to study and conceive new RNN models. Finally, we would like to mention works on the definition of metrics to compare non-linear dynamical systems (Martin (2000); Ishikawa et al. (2018)) although our objective is not exactly the same, as we propose to use dynamical system synchronization theory to improve metric learning on any type of sequential data, whereas these methods have been conceived to work more specifically with structural data.

**Sequence metric learning.** DTW is a classical approach for measuring distances between sequences (Sakoe & Chiba (1978)). Numerous improvements have been brought to the original formulation notably to improve the $k$-nearest neighbor performance (Xi et al. (2006)). Abid & Zou (2018) proposed to learn the DTW parameters that allow to reproduce the Euclidean distances between sequence representations learned with a Sequence-to-Sequence model (Sutskever et al. (2014)). In contrast, Su & Hua (2017) proposed an alternative to DTW, the Order-Preserving Wasserstein (OPW) distance, by viewing the problem of metric learning between sequences as an optimal transport problem regularized to preserve the temporal relationships between the samples, and they solved it with the matrix scaling algorithm. Later (Su & Wu (2019)), the authors reformulated the DTW and OPW distances as parameterized meta-metrics of a single ground metric and proposed an optimization process to learn the metric and the latent alignment with virtual metric learning (Perrot & Habrard (2015)), which reduces the number of constraints. Not only this approach speeds up training but it also outperforms several other metric learning approaches, notably approaches conceived for points generalized to sequences. In comparison, we propose a pure RNN approach similar to Mueller & Thyagarajan (2016) who presented a siamese neural network approach to learn sentence similarities as a $l_1$-norm. In their method, the LSTM network combines the embeddings of the words of the sentence to learn a distance between representations of sentences. Finally, Varior et al. (2016) proposed a siamese convolutional architecture for person re-identification from video data with gates linking parallel layers allowing to accentuate common patterns between both representations. This leads to representations that are more suited to distinguish some pairs of similar or dissimilar images.

In this paper, we introduce an alternative to DTW: a pure neural network approach to sequence metric learning based on the siamese RNN architecture. We propose to enhance the classical Siamese RNN by studying this model from dynamical system point of view, as it has been already done for standard RNN. The disadvantage of DTW-based approaches compared to ours is that it can be

computationally inefficient and non-differentiable, which prevents their possible combination with other gradient-based models.

## 3 SYNCHRONIZING GRU SIAMESE NETWORKS

In this section, we first draw a parallel between the concept of synchronization for dynamical systems and the task of sequence metric learning with siamese RNN. We then justify from a theoretical point of view the introduction of coupling inside the siamese architecture. We finally introduce our main contribution in Section 3.3: a modified Siamese GRU model implementing this coupling.

### 3.1 SYNCHRONIZATION OF CHAOS AND METRIC LEARNING

The concept of synchronization is generally well-understood for time-periodic dynamical systems: this phenomenon is called phase synchronization. However, it is less-known that it can also occur for chaotic dynamical systems (Pecora & Carroll (1990)), that is, systems for which resulting trajectories exponentially diverge for infinitesimally close initial conditions. This practically means that the behavior of such systems can become rapidly unpredictable solely due to small variations of the initial conditions. Common examples of such systems are the double pendulum or the n-body problem. To formalize the concept of synchronization for chaotic systems, Brown & Kocarev (2000) proposed a general definition of it:

**Definition 1** *Let $Z$ be a dynamical system composed of two subsystems $X$ and $Y$ such that:*

$$X : \frac{dx}{dt} = f_1(x, y; t)$$
$$Y : \frac{dy}{dt} = f_2(y, x; t),$$

(1)

*where $x \in \mathbb{R}^{d_1}$ and $y \in \mathbb{R}^{d_2}$ with $d_1, d_2 \in \mathbb{N}$. Let $\phi(z_0)$ be a trajectory of $Z$ with initial conditions $z_0 = [x_0, y_0] \in R^{d_1} \times R^{d_2}$. Finally let $g : X$ (resp. $Y$) $\times \mathbb{R} \to \mathbb{R}^k$ with $k \in \mathbb{N}$, be a measurable property of the subsystems. They are synchronized on the trajectory $\phi(z_0)$ with respect to the property $g$ if there is a time independent function $h : \mathbb{R}^k \times \mathbb{R}^k \to \mathbb{R}^k$ such that:*

$$||h(g(x), g(y))|| = 0,$$

(2)

*where $|| \cdot ||$ is a norm.*

From this definition, it is possible to derive several ways to measure synchronization between two trajectories. Brown & Kocarev (2000) report several slightly different formulations of the synchronization error with the following being the most used. According to them, for identical systems:

$$h(g(x), g(y)) = \lim_{t \to +\infty} (g(x) - g(y)),$$

(3)

where $h$ and $g$ are the same as in Definition 1.

We rewrite this synchronization error for discrete systems in a continuous form by replacing the limit by a comparison of the last element of each trajectory $X$ and $Y$ of length $T$:

$$h(g(X), g(Y)) = X_T - Y_T ,$$

(4)

with $g$ being here a function returning the coordinates of the points. We define $d$ as a distance on discrete dynamical system trajectories derived from the synchronization error by replacing the difference with the Euclidean norm (in definition 1, any norm can be used) to get only positive values:

$$d(X, Y) = ||X_T - Y_T||_2.$$

(5)

However, RNNs are dynamical systems, and the output sequences are trajectories. Thus, learning a Euclidean distance with a siamese RNN is equivalent to trying to synchronize the output sequences of the two sub-networks of the siamese network for similar pairs.

While being intuitive and suitable for metric learning, the metric of Equation 5 measures synchronization only at one point in time, which forces the system to achieve synchronization at this precise point.

This is called dead-beat synchronization, synchronization in a finite number of steps (De Angeli et al. (1995)). Even if at first sight, this seems not really different from computing distance on input sequence representations, synchronization could actually be assessed at several samples of the sequence and even continuously. We will now specify what type of synchronization siamese RNNs are able to achieve and under which conditions.

## 3.2 COMPLETE SYNCHRONIZATION OF COUPLED IDENTICAL SYSTEMS

A special case of Definition 1 is when $f_1$ and $f_2$ are the same function, the dynamical systems share the same parameters, and they are said to be *identical*. Analogously, the two (or more in case of triplet inputs) sub-networks of a siamese network also share the same parameters; only input sequences differ (Bromley et al. (1994)). To simplify, we will first study the case where the dynamics of the RNNs are solely driven by its initial condition (the initial hidden state) and where no input sequence is given (i.e. a sequence of 0s). We obtain what is called the *dynamical system induced* by the RNN. In this case, only the initial conditions differ and the sub-networks are identical dynamical systems. According to experiments conducted by Laurent & von Brecht (2016), dynamical systems induced by RNN exhibit a chaotic behavior. Under what conditions identical systems can synchronize? Consider now the following two identical systems:

$$X : \frac{\mathrm{d}x}{\mathrm{d}t} = f(x; t) + C(y - x)^T$$
$$Y : \frac{\mathrm{d}y}{\mathrm{d}t} = f(y; t) + C(x - y)^T, \tag{6}$$

where $x, y \in \mathbb{R}^n$ and $C$ is a coupling matrix in $\mathbb{R}^{n \times n}$. This type of coupling is called *diffusive* because it will dissipate the dynamics of each sub-system (Boccaletti et al. (2002)). $X$ and $Y$ are bidirectionally coupled systems[1]. Fujisaka & Yamada (1983) showed that the system described by Equation 6 can achieve complete synchronization if $C$ is a multiple of the Identity matrix and a constant $c$ which verifies the following condition:

$$c > \frac{1}{2} \lambda_L, \tag{7}$$

where $\lambda_L$ is the largest Liapunov exponent of the system. The Liapunov exponents quantify the sensibility of a system to initial conditions: if it has one positive Liapunov exponent, predictability of its behavior becomes impossible beyond a certain time horizon; it is therefore chaotic (Strogatz (2018)).

But the trajectories of RNNs are most of the time also influenced by external inputs: the input sequence. In this case, the dynamics of the RNNs are mostly driven by these external inputs (Laurent & von Brecht (2016)) and RNN starting from different initial conditions but given identical input sequence will see their trajectories synchronize, i.e. the hidden states become the same after a few steps. Coupling is in this case not necessary to achieve synchronization: regarding metric learning, siamese LSTM actually works without coupling (Mueller & Thyagarajan (2016)). However, coupling could allow to enforce lower distances with sequences that have similar dynamics but are composed of quite different data, i.e. so-called hard positive samples, and even to force the synchronization regardless of the input pair. Hard positive samples are samples that are factually belonging to one class, by the label, but lie close to the decision boundary or even beyond in terms of distance (conversely for hard negative samples). They are particularly studied by the metric learning community to improve the convergence and performances of metric learning models Schroff et al. (2015).

We showed in this section the motivation and aim of implementing coupling into the siamese RNN architecture. Indeed, the induced dynamics of GRU and LSTM are chaotic and, in this case, coupling allows their complete synchronization as the sub-networks of a siamese network share the same weights and represent thus identical dynamical systems. When given input sequences, the dynamics of GRU and LSTM are mostly driven by these external inputs. In this case, while not being critical to achieve synchronization (and therefore low distances between similar elements), coupling could facilitate bringing similar inputs closer, particularly for hard positive pairs.

---

[1] Coupling can also be directional, only one system influences the other: they are called in this case drive-response systems.

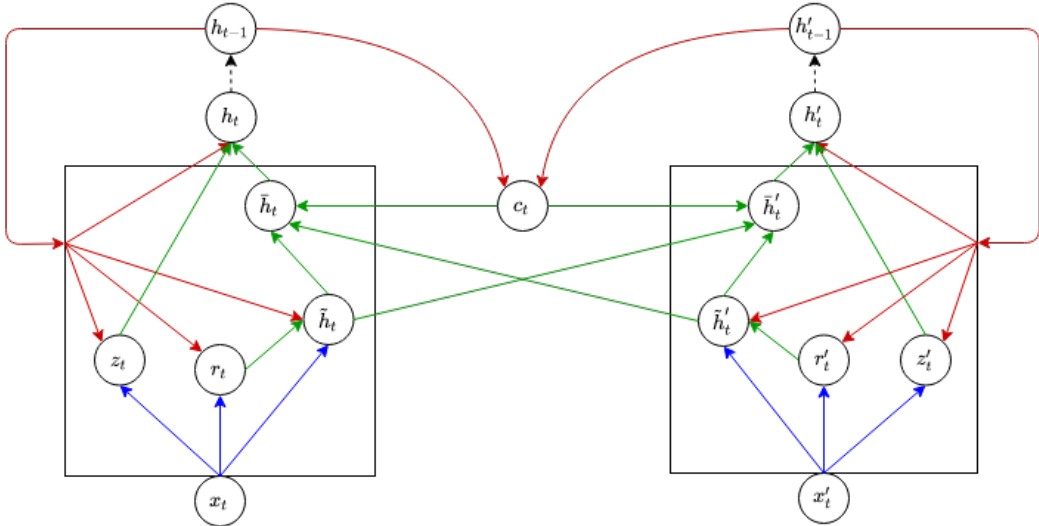

Figure 1: Schema of the CGRU architecture. Blue arrows represent information coming from the input at time $t$, red ones are for the hidden state and green ones for transmissions between the gates.

### 3.3 COUPLED GRU

We present a new neural network model that directly implements coupling within a siamese RNN architecture. From a machine learning perspective, this coupling needs to be trainable such that the network *learns* to achieve synchronization for similar inputs and stay desynchronized for different ones. We propose to apply the coupling by means of two new gates inside the GRU architecture which we call in the following CGRU (Coupled Gated Recurrent Unit). We chose to use GRU and not LSTM (Hochreiter & Schmidhuber (1997)) because the operation of a GRU is simpler (i.e. defined by fewer equations) while showing comparable performance in general (Chung et al. (2014)). The following equations describe the modifications brought to the architecture. Update, Reset and New gates are not modified. Let us notate $h'_{t-1}$ and $\tilde{h}'_t$ the states coming from the second sub-network (see Figure 1):

$$\text{Hidden state: } h_t = (1 - z_t)\bar{h}_t + z_t h_{t-1} \tag{8}$$
$$\text{Update: } z_t = \sigma(W_{iz}x_t + b_{iz} + W_{hz}h_{t-1} + b_{hz}) \tag{9}$$
$$\text{Coupled New State: } \bar{h}_t = (1 - c_t)\tilde{h}_t + c_t \tilde{h}'_t \tag{10}$$
$$\text{Coupling: } c_t = \sigma(W_{hc}(h_{t-1} + h'_{t-1}) + b_{hc}) \tag{11}$$
$$\text{New State: } \tilde{h}_t = \tanh(W_{i\tilde{h}}x_t + b_{i\tilde{h}} + r_t(W_{h\tilde{h}}h_{t-1} + b_{h\tilde{h}})) \tag{12}$$
$$\text{Reset: } r_t = \sigma(W_{ir}x_t + b_{ir} + W_{hr}h_{t-1} + b_{hr}). \tag{13}$$

The Coupling gate $c_t$ (see Equation 11) serves the same purpose as $z_t$ and $r_t$, controlling the information flow and is thus computed in a similar fashion, but only from the hidden states. This forces the model to apply the coupling on the new content to be added at time $t$ solely based on the previous inputs. Then, in Equation 10, $\tilde{h}_t$ and $\tilde{h}'_t$ are combined similarly as $\tilde{h}_t$ and $h_{t-1}$ are combined in the original GRU architecture. This prevents $\bar{h}_t$ and subsequently $h_t$ from exploding and saturate the gates. Finally, in Equation 8, $\bar{h}_t$ replaces $\tilde{h}_t$: the New state has been replaced by a coupled version of both New states of the siamese GRU. Several possibilities exist to implement this coupling. The idea behind this proposal is to alter as little as possible the GRU architecture and to stay close to the original purpose of each equation. Indeed, the addition of the coupling already greatly modifies the information flow inside the GRU and the gradient flow during training (see the supplementary material for the complete differentiation of CGRU), and RNNs are known to be difficult to train. Therefore, by staying relatively close to the original model, a rigorous comparison is simpler, and the impact of the actual coupling can be studied more reliably. In fact, if $c_t$ is a vector of norm equal to zero, each sub-network is exactly a GRU. This suggests to initialize the coupling weights with very

small values and to accentuate the decay. In this way, an increase of the norm of $W_{hc}$ during training would signify that coupling is useful. Another interesting configuration of the coupling weights is when they are all equal to 0.5: in this configuration, the Coupled New States are the same, and the distance between the outputs will become null. That means, theoretically, this approach can make close any pair of input sequences, especially hard-positive samples.

## 4 EXPERIMENTS

### 4.1 EXPERIMENTAL SETUP

We experiment the CGRU architecture on the dataset UCI HAR[2] (Anguita et al. (2013)), a dataset of 6 activities[3] containing 9 features: total acceleration, body acceleration and angular velocity on 3 axes. The sequences have a length of 128. We chose this dataset because it has been extensively used by the activity recognition community and provides at the same time real data and a simple and well defined benchmark to study the behavior of CGRU and Siamese GRU (SGRU). Moreover, several activities should look very similar (e.g. three variants of walking or standing and sitting), and it should make the dataset harder to process for metric learning algorithms. Finally, *walking* or *running* exhibits dynamic components which could be differently processed by CGRU and SGRU. No further preprocessing has been applied. The features are globally centered and the standard deviations oscillate between 0.1 and 0.4. We kept the train-test split proposed by the authors of the dataset: there are 21 users in the training set and we therefore performed a 7-fold validation, leaving each time 3 different users out. Finally, the training set comprises 7352 sequences and the testing set 2947.

We compared SGRU and CGRU on learning the metric describes in Equation 5 using the same hyperparameters for both models. We used the same architecture for both networks, a one-layer network with 20 neurons and an initial learning rate of 0.001 which we decreased by a factor of 0.5 after 10 epochs if training loss does not improve. The training is stopped based on the accuracy on the validation set (early stopping). We chose to use structural loss (Yang et al. (2018)) to train the model since it is a recent loss working on distances and not embeddings. It combines a local term similarly to the n-pair loss (Sohn (2016)) but emphasizes the weights on the hard positive samples, and a global term to improve the generalization. We used the same hyperparameters for the loss as in the original paper. The full equations of the loss are available in the second section of the supplementary material. Regarding the batch size, each batch contains 6 elements of each class (total 36) to form a valid batch for structural loss. We applied a general weight decay with a factor of $10^{-4}$. A stronger weight decay was applied on the coupling by adding 1% of the coupling weight norm to the loss, which seems to slightly improve the performances. The gradient was clipped according to (Pascanu et al. (2013)) to a norm of 6. The coupling weights were initialized with a normal distribution having a mean of 0 and standard deviation of 0.1. The purpose of this initialization is to make the model start its training close to the behavior of an SGRU, with a very weak coupling and to let it increase during training. Our implementation was done in Python and CUDA with Pytorch (Paszke et al. (2019)).

### 4.2 RESULTS

#### 4.2.1 STUDY OF THE COUPLING WEIGHT NORM

We first analyze the evolution of the coupling weight norm during training. The coupling is initialized very low which theoretically makes it behave nearly as an SGRU at the beginning of training. We can observe in Figure 2a that the norm increases quickly during the first 20 epochs and more than doubles. Red points indicate the iterations where validation accuracy increased. This correlation thus indicates that the overall generalization is improving with the increase of coupling strength. In Figure 2b, we present the evolution of the loss on the train dataset in terms of the coupling weights on which we compute a linear regression of the first part. We can observe an almost linear relationship between the increase of the norm and the decrease of the loss suggesting again that the coupling is helpful to the model. Those figures also show that the convergence during training is rather smooth.

---

[2] https://archive.ics.uci.edu/ml/datasets/human+activity+recognition+using+smartphones

[3] walking, walking upstairs, walking downstairs, sitting, standing, laying

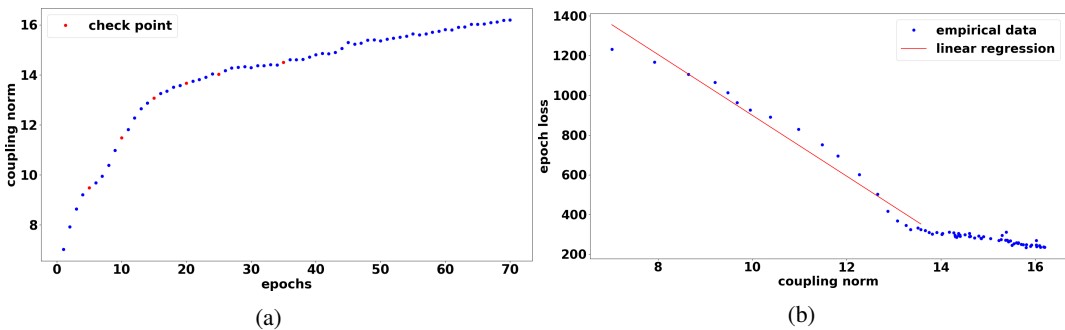

(a)                            (b)

Figure 2: In figure (a), evolution of the norm of the coupling gate weights during training on UCI HAR dataset, a red point indicates an increased accuracy for the validation set. On figure (b), Epoch loss in terms of coupling norm during training on UCI HAR dataset. The first part (rapid decrease) has been approximated with a linear regression. The correlation coefficient is -0.984.

| Algorithms | Accuracy | F1 score Macro | MAP |
|---|---|---|---|
| Siamese GRU | 0.835±0.068 | 0.827±0.074 | 0.711±0.016 |
| Coupled GRU | **0.913±0.055*** | **0.916±0.054*** | **0.900±0.043*** |

(a) Validation results (21 fold average). An asterisk means a significant result with a threshold of 1%

| Algorithms | Accuracy | F1 score Macro | MAP |
|---|---|---|---|
| RVSML (OPW) (Su & Wu (2019)) | 0.597 | 0.568 | 0.438 |
| RVSML (DTW) (Su & Wu (2019)) | 0.698 | 0.687 | 0.437 |
| Siamese GRU | 0.782±0.041 | 0.781±0.044 | 0.633±0.152 |
| Coupled GRU | **0.885±0.014*** | **0.887±0.014*** | **0.899±0.01*** |

(b) Test results, average of 5 run for the neural networks approaches.

Table 1: Results on UCI HAR

### 4.2.2 CLASSIFICATION PERFORMANCE ON UCI HAR

We now present classification results on UCI HAR using 3 metrics : accuracy, F1 score Macro averaged and Mean Average Precision (MAP), similarly to Su & Wu (2019). The first two are computed by classifying the test samples using 1-nearest neighbor from training samples. The MAP is computed by querying the training set with a test sequence to retrieve all training samples of the same class. The value is averaged for all test sequences. This metric shows the ability of the algorithm to bring close every sequence of each class and not just few references to be used as nearest neighbors. The validation results are presented in Table 1a. We observe an improvement of CGRU over SGRU of about 8% points for accuracy and F1-score, and an improvement of 19% points for the MAP. On the test set, we compared our approach with Regressive Virtual Sequence Metric Learning (RVSML) (Su & Wu (2019)), with OPW and DTW distances. We chose to compare with this approach because it recently outperformed several other metric learning approaches (e.g. Large Margin Nearest Neighbors (Weinberger & Saul (2009)), Regressive Virtual Metric Learning (Perrot & Habrard (2015)) etc.) although not all were specifically adapted for sequences, it is not based on neural networks and uses alignment-based distances. We keep the hyperparameter values recommended by the authors. The test results are presented in Table 1b. Here again we observe that CGRU outperformed SGRU with a notable improvement of 10% points of the accuracy and F1-score. Both neural network approaches clearly outperformed RVSML, especially the OPW variant. This can be, among other factors, attributed to a weak capacity to distinguish similar activities such as *standing* and *sitting*. On the other hand, *standing* was recognized perfectly by both RVSML variants. We also remark that they reach MAP values of the same order as in the original paper (around $0.4 \sim 0.45$) despite the fact that the data are of a completely different nature (signal instead of images). This could suggest that these approaches reached some kind of saturation whereas CGRU and SGRU are able to achieve much higher values.

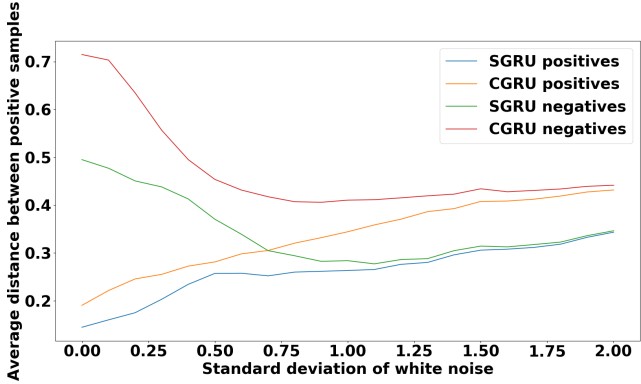

Figure 3: Evolution of the average distance between the positive/negative samples for SGRU and CGRU on more and more noisy test sets.

### 4.2.3 Performances on hard positive samples

We made the hypothesis that CGRU could perform better on hard positive samples due to coupling theoretically being able to bring close any input pair of sequences: with enough coupling, both networks can output the same sequence whatever of the input pair. We propose to verify this by observing the evolution of the average distance between the positive samples and between the negative samples when white noise is gradually added to the feature sequences of the testing set. The standard deviation of the noise goes from 0 to 2. We also note that both models were trained up to comparable validation accuracy. The results are presented on Figure 3. We observe that the curves for both models evolve similarly with positive distances gradually increasing with the noise. The negative and positive curves join the moment too much noise is added and the sequences become indistinguishable. CGRU produces slightly higher distance average than SGRU but is able to maintain higher margins more longer: up to 1.75 units of standard deviation compared to about 1.25 for SGRU. This shows that, as theoretically possible with the coupling, CGRU better discriminate the hard samples. The third section of the supplementary material extends this argumentation by analyzing average distance to nearest neighbors.

## 5 Conclusion and Perspectives

We presented a new framework for sequence metric learning based on dynamical system synchronization theory. We drew a parallel between synchronized trajectories and output sequences of siamese recurrent neural networks produced from similar input pairs. After characterizing Siamese GRU as identical chaotic systems, we showed the contribution of introducing coupling inside the siamese architecture to achieve synchronization more easily and to increase the capacity of the network to bring closer the embedding of some similar input pairs, especially hard positive samples. This coupling was implemented through a new gate inside the Siamese GRU architecture which allows the network to mix the new content of both sides of the siamese network. Our experiments showed that the Siamese GRU architecture benefits from the coupling, can be smoothly trained with it and fits well with recent complex metric learning losses such as structural loss. CGRU proved to be outperforming SGRU with the same architecture on an activity recognition benchmark and was able to maintain a higher margin for hard samples.

The study of sequence metric learning with synchronization opens several perspectives especially to design new forms of metrics by taking inspiration from the literature of synchronization criteria (Schiff et al. (1996); Kreuz et al. (2007)) though non-differentiability, notably when those metrics use mutual neighbors which are computed using explicitly the time steps, has to be overcome in many cases thanks to an attention mechanism, for example. The coupling itself could be tuned with theoretical contributions (Brown & Rulkov (1997)). One drawback of the proposed architecture is that each pair has to be passed through the network instead of just computing once each representation and then the distance for each pair. This could be balanced by the use of virtual metric learning during training. Finally the coupling allows to bring any pair of inputs close to one another if sufficiently

strong and could be use as an indicator in weakly supervised settings to invert the equivalence constraint of some pairs dynamically if the network is forcing the synchronization too much.

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
