# OpenReview forum: "Sequence Metric Learning as Synchronization of Recurrent Neural Networks"
_ICLR.cc/2021/Conference — Reject_

### Official Review · AnonReviewer4 · 2020-10-17
**Not strong contribution with seemingly wrong paper highlight**

**Rating:** 3
**Confidence:** 3

**Review:**

#### Summary:
Drawing inspiration from dynamic systems, the paper proposes a novel architecture that couple sequences. Such a system has easiness to bring two instances arbitrarily close and authors have shown the superiority of the approach over an action recognition dataset; but the results seem to far from state of the art on the dataset (see questions section). The authors also recognize that currently such systems need to calculate each pairs (can't be cached due to coupling) at inference time, which is slow.

The main issue I found in this paper is its presentation. The authors claim to draw inspirations from dynamic systems (e.g. the important notion of  synchronized trajectories) in the abstract/introduction, cite related work and introduce a formal definition (with some errors, see section minor issues). However, the metric that ends up being used in eq(5) is a metric function of two RNN end states, which is common and wouldn’t help to highlight authors’ contributions.

Another concern is that the authors have tested on only one dataset where the claimed results are quite below state of the art on the dataset (see questions section). There are two baselines implemented (one very recent approach), however no simple baseline or other datasets to further support the approach.


Finally, after reading the paper, I think the paper proposes a new neural architecture for similarity learning rather than focusing on metric learning. In this aspect, the paper misses references to papers in the area.


#### Pros:
Drawing inspiration from dynamic systems, the paper proposes a novel architecture that is not only dynamically involved with timesteps but is also coupled (between two instances), which has the capability of bringing two arbitrary sequences close. The authors have shown the superiority of the proposed approach over several metric learning baselines including the recently proposed RVSML (Su and Wu, 2019) over an action recognition dataset.

#### Questions:
Clarification question: Given the architecture, the methods can only be applied to sequences that have the same number of timesteps, is that correct?

The approach seems to be far from the state of art: In Provable Defenses against Adversarial Examples via the Convex Outer Adversarial Polytope [Wong and Kolter, 2018], the baseline network investigated achieves 5% error rate; for papers more focused on performance, Human activity recognition with smartphone sensors using deep learning neural networks [Ronao and Cho 2016] archives accuracy 95.75%. Why is there such a discrepancy between the results in the paper and state of the art results please (as it is more convincing to build upon state of the art results when possible).

#### Minor Issues:
In Definition 1, the function two subsystems are synchronized if there exists a time-independent g (instead of a time dependent g as written). In fact, one can always find a trivial time dependent g satisfying the equation (by the way, it is written correctly with time independence in Brown & Kocarev (2000))

#### Minor suggestions:

Modifying neural architecture to capture better similarity has numerous works, one notable example in NLP is for example: A Decomposable Attention Model for Natural Language Inference [Parikh et al. 2016]. I would recommend including similar references in the paper to better situate the paper’s contribution.

The loss function is missing in the paper, so leading some missing details for training.

Finally, I want to detail my argument to change the paper’s highlight on synchronization, which ends being a distance between two RNN end states. First, this is very far from the formal definition in eq(1) with a quite weak link; secondly, it is one of the most commonly used metrics in NLP/CV applications (and very probably other areas that I am much less familiar with). Given that an ICLR main paper is only about 8 pages, I strongly recommend authors to reconsider the paper structure to highlight its main contributions.

---

### Official Review · AnonReviewer3 · 2020-10-28
**Good idea, but experimentation is weak**

**Rating:** 4
**Confidence:** 5

**Review:**

# Summary

This work concerns the metric learning between sequences using RNNs. The paper notices the similarity between a dynamical system and an RNN. Then it demonstrates that learning a pair of siamese RNNs is similar to learning synchronization between two subsystems of a dynamical system. Finally, the paper proposes to introduce coupling between the two RNNs in order to improve synchronization.

The paper conducts experiments comparing the siamese GRUs with the proposed couples siamese GRUs.


# Quality

While this is an interesting idea to explore, I am not entirely convinced about its value. The main claim of the paper seems to be that the baseline siamese GRU exhibits chaotic behavior while the proposed architecture does not. The former has to be tested experimentally. I would like to see an experiment that shows this chaotic behevior at least qualitatively. Secondly, the proposed architecture intermixes two sequences. Therefore, it is not possible to learn embeddings.

The experimental section is very scarce. The only dataset used in the paper, UCI HAR, is very small. Furthermore, there are not enough benchmarks for the metric learning on this dataset. The results reported here are not comparable with the supervised methods (~90-95%). This work needs more experiments on datasets with several published results.

The section 4.2.1 seems to contain a logical error. The linear relation between the norm and the loss cannot show that the coupling helps or hurts the model. Instead, a model with coupling should be compared to a model without coupling. Would it be better to plot separately the norm for positive samples and the negative samples? If I understand correctly, the first is supposed to decrease and the second is supposed to increase. There is more chance to see a negative pair, therefore you observe that the norm increases.

The section 4.2.3 aims to test the model for the hard positive samples. I can see several problems with the proposed approach to test this. Firstly, the absolute distance is meaningless here. I propose to normalize both curves. Secondly and most importantly, the average metrics cannot test the performance for the hard positives. Such metrics would be dominated by the majority of "easy positives". Therefore, I request a more formal description of what is a "hard positive" and an experiment  that better tests the claim.

# Clarity

In general, the paper is hard to read. The logical flow of the paper is hard to follow. Some important aspects are only referenced in other papers or skipped through. Many typos hinder the understanding sometimes.

The introduction seems to claim "an improvement over a classical GRU". In fact, the paper proposes an improvement to the *siamese* GRUs.

Despite the fact that the proposed idea is quite simple, it was hard to follow the paper. Sections 3.1 and 3.2 are too general. This creates discrepancy between Sections 3.1-2 and 3.3. I don't see the point of introducing so general description of the synchronization. It would be clearer to define this concept for the specific case at hand.

The abuse of notation made Definition 1 look nearly incorrect. I recommend either carefully introduce the specifics here (like in the original paper by Brown & Kocarev) or stick to the common mathematical notation. More specifically:
- Definitions for "X", "Y", and "X (resp. Y)"
- Definition for g(x)

It is easy to loose a thread when reading the section 3.1. I would recommend to start with the siamese RNNs and then demonstrate the connection to the dynamical system, not other way around.

The loss used in the paper has to be explicitly spelled out (perhaps, in the Appendix).

# Originality

The paper is sufficiently original.

# Significance

The paper has a potential to be have high significance. Unfortunately, the experimentation is too limited. The claims made in the paper are not sufficiently tested. Then, the performance is tested only on one dataset against a weak baseline. The presentation needs to be improved too (as noted above).

# Conclusion

The idea is interesting, but the experimentation is very weak.

Specific requests for improvement:

- Experiments on more datasets
- Compare to the published metric learning literature
- Test the claim that the baseline is chaotic
- Define hard positive samples and test the model on them
- Improve Definition 1
- Conduct ablation experiments

# Some of typos

- Fix \citep vs \citet
- "further lose temporal dependency"
- "in term of synchronization"
- "system theory, an important result being ..."
-  "recent approach <...> studied"
- "further introduce"
- "h and g are the same as in Equation 1" -- there are no h and g in Equation 1
- "RNN are dynamical systems" -> "RNN is.."
- "RNN are known" -> "RNNs are" or "RNN is"
-  "3 axis" -> "3 axes"
- "size of 128" -> "length of 128"
- "a initial learning rate" -> "an initial"
- "a SGRU" -> "an SGRU"

# Update

Thank you for the rebuttal. The writing was improved and it is easier to read the paper. Nevertheless, the experimentation remains weak. I increase the rating by one point.

---

### Official Review · AnonReviewer1 · 2020-10-31
**The authors describe coupling view of the temporal LSTM/GRU style networks, suggesting modifications to the GRU called SGRU that offers improved results on UCI activity recognition against SGRU and RVSML style approaches.**

**Rating:** 6
**Confidence:** 3

**Review:**

I liked the formulation and motivation of the paper, explaining the sequence metric learning problem  and drawing parallel between synchronized trajectories produced by dynamical systems and the distance between similar sequences processed by a siamese style recurrent neural network. The authors propose modification the siamese recurrent network setting called classical Gated Recurrent Unit architecture (CGRU). The premise being two identical sub-networks, two identical dynamical systems which can theoretically achieve complete synchronization if a coupling is introduced between them. The authors describe how this model is able to simultaneously learn a similarity metric and the synchronization of unaligned multi-variate sequences in a weakly supervised way with the coupling demonstrating performance of the siamese Gated Recurrent Unit (SGRU) architecture on UCI activity recognition dataset (mobile data).

The increase in norm with the epochs shows overall generalization is improving with the increase of coupling strength and an almost linear relationship between the increase of the norm and the decrease of the loss suggesting again that the coupling is helpful. Computing accuracy, F1 score Macro averaged and Mean Average Precision (MAP),similarly to Su & Wu (2019) also shows improvement against SGRU and RVSML.  The authors mention a drawback of the proposed architecture is that each pair has to be passed through the network instead of just computing once each representation and then the distance for each pair and that this could be balanced by the use of virtual metric learning during training.

The improvements points in the paper are
- Comparing against more recent work on the activity recognition such as  Learning Discriminative Virtual Sequences for Time Series Classification
- using more comprehensive evaluation (rather than a single data set)
- will also be good to expand on the virtual  metric learning during training as passing each pair can increase the training complexity for large datasets

---

### Author Response · Authors · 2020-11-24
**Response to the reviewers (first part)**

First of all, we would like to sincerely thank the reviewers for the detailed and constructive reviews of our work, notably to improve the experimental section and the organization of the paper.
The general orientation of our paper was to propose a new point of view on the sequence metric learning problem, then from here to derive an architecture and to analyze some of its interesting properties.
As noticed by several reviewers, there was typos and notation inconsistencies in Definition 1 which we corrected.
The loss we used in this paper comes from [1]. We added a detailed description of it in Section 2 of supplementary material.

**Reviewer 1:**

Thank you very much for your comments on our work.

1. “Comparing against more recent work on the activity recognition such as Learning Discriminative Virtual Sequences for Time Series Classification”
“will also be good to expand on the virtual metric learning during training as passing each pair can increase the training complexity for large datasets”

Thank you for the suggestion, we will take a look at this work and see where it fits compared to ours. Indeed we already started to investigate virtual metric learning as a solution to overcome the limitations imposed by mixing both sequences. However, preliminary experiments tell us that it requires lots of regularization and a careful construction of the virtual sequences due to the asymmetry created by passing a “true” sequence and a virtual one. Therefore, further work is required.

2. “using more comprehensive evaluation (rather than a single data set)”

Thank you for your suggestion of improvement of our experimental section. Indeed the paper would gain in strength with experiments on more datasets.

**Reviewer 2:**

Thank you very much for your comments on our work.

1. “The main claim of the paper seems to be that the baseline siamese GRU exhibits chaotic behavior while the proposed architecture does not. The former has to be tested experimentally. “

We are sorry for the lack of clarity of the paper on that point but this is not our claim. This claim is made in [2] and observed experimentally for LSTM (page 4). We used this paper to clarify the behavior of the Siamese GRU regarding dynamical systems synchronization theory (uncoupled chaotic identical system) and therefore to support the idea of introducing coupling inside the architecture. We then adapt this idea of coupling coming from dynamical system theory to machine learning and a Siamese GRU architecture. This is the main contribution of our paper presented in section 3.3.

2. “Secondly, the proposed architecture intermixes two sequences. Therefore, it is not possible to learn embeddings.”

This is correct and while it can be seen as a drawback, our approach also seeks to distance itself from representation learning strategies and to propose a new deep learning approach for sequence metric learning.

3. "The experimental section is very scarce. The only dataset used in the paper, UCI HAR, is very small. Furthermore, there are not enough benchmarks for the metric learning on this dataset. The results reported here are not comparable with the supervised methods (~90-95%). This work needs more experiments on datasets with several published results."

Thank you very much for your suggestions to strengthen our experimental section. Indeed our architecture would require to be tested on several datasets to fully prove (or not) that it is better than SGRU. We did not try to compare with supervised classification methods on this dataset which indeed achieve higher results. Our goal was rather to demonstrate that CGRU could achieve better accuracy than SGRU using the same hyperparameters. For this reason, we only provided comparison with metric learning approaches.

4. “The section 4.2.1 seems to contain a logical error. The linear relation between the norm and the loss cannot show that the coupling helps or hurts the model. Instead, a model with coupling should be compared to a model without coupling. Would it be better to plot separately the norm for positive samples and the negative samples? If I understand correctly, the first is supposed to decrease and the second is supposed to increase. There is more chance to see a negative pair, therefore you observe that the norm increases.”

In section 4.2.1, we are dealing with the norm of the coupling weights (the model parameters), during training, not the distances between the samples. The weights are initialized very close to zero, in this situation, the architecture behaves nearly as a Siamese GRU (see Eq (10) and Eq (11)), we then observe that this norm (of the coupling parameters) increases during training following the validation accuracy which suggests that the couping parameters contribute to improve our classification accuracy on UCI dataset.

---

> ### Author Response · Authors · 2020-11-24
> **Response to the reviewers (second part)**
>
> 5. “The section 4.2.3 aims to test the model for the hard positive samples. I can see several problems with the proposed approach to test this. Firstly, the absolute distance is meaningless here. I propose to normalize both curves. Secondly and most importantly, the average metrics cannot test the performance for the hard positives. Such metrics would be dominated by the majority of "easy positives". Therefore, I request a more formal description of what is a "hard positive" and an experiment that better tests the claim..”
>
> Thank you very much for your remark on normalization, we will take it into account. From our point of view, hard samples are samples close to the decision boundary, factually (by the label so to say) belonging to one class but by the distance (at some moment during learning or testing) being closer to samples from another class. Depending on the sampling strategy, this hardness can be quantified by several means (triplets for those the distances between the anchor and the positive and negative samples are close can be considered hard). We actually agree with your comment, however as the standard deviation of noise increases, we can argue that there are only “hard samples” since noise is applied on each sequence. This is notably attested by the fact that the average distances between the positive and negative samples are converging. Nevertheless, we will precise in the paper our definition for “hard samples”.
>
> 6. "Despite the fact that the proposed idea is quite simple, it was hard to follow the paper. Sections 3.1 and 3.2 are too general. This creates discrepancy between Sections 3.1-2 and 3.3. I don't see the point of introducing so general description of the synchronization. It would be clearer to define this concept for the specific case at hand."
>
> Thank you for all suggestions. In section 3.1 we wanted to show that a link could be made between synchronization and metric learning, we therefore needed to introduce synchronization and synchronization error. In section 3.2, we wanted to justify the introduction of coupling and we therefore also needed to introduce the complete synchronization case before presenting in 3.3, the core contribution of the paper. We tried to make this clearer in the paper.
>
> 7. “Conclusion”
>     * "Experiments on more datasets" :
>       Thank you very much for your suggestion which we discussed above.
>     * "Compare to the published metric learning literature" :
>       Thank you for your suggestion. We provided a comparison with RVSML [4] which is quite recent but more comparisons could indeed be added to reinforce the interest of our approach.
>     * "Test the claim that the baseline is chaotic" :
>       We discussed this remarked above.
>     * "Define hard positive samples and test the model on them" : Thank you for your suggestion, we also discussed this remark above and tried to improve the paper accordingly.
>     * "Improve Definition 1"
>       Thank you very much we reworked definition 1 accordingly.
>     * "Conduct ablation experiments"
>       Thank you for your suggestions of improvement of our experimental section. The difference between SGRU and CGRU is just the introduction of the coupling. Otherwise, both architectures were tested with the same setup. Moreover, this coupling is nullified for a certain configuration of the parameters (if W_hc and b_hc are null matrices) which makes CGRU actually a SGRU.
>
> 8. “Some of typos”
>
> Finally, thank you for having highlighted some typos which we corrected.

---

> > ### Author Response · Authors · 2020-11-24
> > **Response to the reviewers (third part)**
> >
> > **Reviewer 3:**
> >
> > Thank you very much for your comments on our work.
> >
> > 1. “The main issue I found in this paper is its presentation. The authors claim to draw inspirations from dynamic systems (e.g. the important notion of synchronized trajectories) in the abstract/introduction, cite related work and introduce a formal definition (with some errors, see section minor issues). However, the metric that ends up being used in eq(5) is a metric function of two RNN end states, which is common and wouldn’t help to highlight authors’ contributions.”
> > and “First, this is very far from the formal definition in eq(1) with a quite weak link; secondly, it is one of the most commonly used metrics in NLP/CV applications”
> >
> > Thank you for your suggestions to improve the clarity of the paper. In Definition 1 or Equation (3), we consider continuous time dynamical systems and infinite trajectories. Eq (5) is a discrete version for finite trajectories otherwise each element stated in definition 1 still holds: g is the coordinates (a property of the system), h the difference and a norm appears. We wanted to make the analogy between metric learning on sequences with recurrent neural networks and synchronization. This serves to justify the introduction of our main contribution in section 3.3.
> > Regarding the simplicity of the final learned metric, more complex synchronization error equations could be use to produce more complex metrics (possibly on the whole output sequences). However, since this one works well with identical systems (in its continuous form) and is ubiquitous in metric learning (in the discrete form), we had no reasons to introduce something else much more elaborated at this level and this is not a main contribution of this paper. We tried to clarify this in the paper so to avoid confusion.
> >
> >
> > 2. “Given the architecture, the methods can only be applied to sequences that have the same number of timesteps, is that correct?”
> >
> > This is correct however this drawback can be mitigated with padding or/and resampling.
> >
> > 3. “Why is there such a discrepancy between the results in the paper and state of the art results please”
> >
> > This paper was not focused on performances and especially beating the approaches you mention which are supervised classification approaches. Our goal was rather to demonstrate that CGRU could achieve better performances than SGRU with the exact same architecture, at least in this case. That is why we only provided comparison with metric learning approaches.
> >
> > 4. "In Definition 1, the function two subsystems are synchronized if there exists a time-independent g (instead of a time dependent g as written). In fact, one can always find a trivial time dependent g satisfying the equation (by the way, it is written correctly with time independence in Brown & Kocarev (2000))"
> >
> > Thank you for your remark, we have reworked definition 1 accordingly.
> >
> > 5. “Modifying neural architecture to capture better similarity has numerous works, one notable example in NLP is for example: A Decomposable Attention Model for Natural Language Inference [Parikh et al. 2016]. I would recommend including similar references in the paper to better situate the paper’s contribution.”
> >
> > Thank you very much for your suggestion, we will look into this literature and see where our work fits in it. We mentioned in the state of the art the approach of Varior et al. [3] which seems quite close in the realization although not for sequences.
> >
> > 6. "The loss function is missing in the paper, so leading some missing details for training. "
> >
> > Thank you very much for your remark, we added it in the supplementary material, Section 2.
> >
> > 7. "Finally, I want to detail my argument to change the paper’s highlight on synchronization, which ends being a distance between two RNN end states. First, this is very far from the formal definition in eq(1) with a quite weak link; secondly, it is one of the most commonly used metrics in NLP/CV applications (and very probably other areas that I am much less familiar with). Given that an ICLR main paper is only about 8 pages, I strongly recommend authors to reconsider the paper structure to highlight its main contributions."
> >
> > The main contribution of this paper is presented in Section 3.3. Section 3.1 justifies the analogy that can be made between synchronization and metric learning. Section 3.2 tries to justify the introduction of coupling which is done by modifying the Siamese GRU architecture in section 3.3. Following your suggestions, we tried to make this clearer in the paper.

---

> > > ### Author Response · Authors · 2020-11-24
> > > **References**
> > >
> > > [1] Yang, X., Zhou, P., & Wang, M. (2018). Person reidentification via structural deep metric learning. IEEE Transactions on Neural Networks and Learning Systems, 30(10), 2987-2998.
> > >
> > > [2] Laurent, T., & von Brecht, J. (2016). A recurrent neural network without chaos. arXiv preprint arXiv:1612.06212.
> > >
> > > [3] Varior, R. R., Haloi, M., & Wang, G. (2016, October). Gated siamese convolutional neural network architecture for human re-identification. In European conference on computer vision (pp. 791-808). Springer, Cham.
> > >
> > > [4] Su, B., & Wu, Y. (2019, May). Learning Distance for Sequences by Learning a Ground Metric. In International Conference on Machine Learning (pp. 6015-6025).

---

### Decision · Program_Chairs · 2021-01-07
**Final Decision**

**Decision:**

Reject

**Comment:**

The paper proposed to learn a sequence metric by sharing a memory cell between two LSTMs that run on pairs of sequences. I found the idea quite interesting, but I (and the reviewers) found the inspiration and the analogy from a dynamical systems perspective a unclear, unconvincing and maybe not even necessary to the core essence of the method that was proposed.

 The reviewers appreciated the improvement in clarity over the course of the review, but felt that there was still some more distance to cover. In addition, the results were not of a high enough quality to lend support to the success of the method and the experimental section needs more work making it not ready for ICLR acceptance at this time.